# Interpretable Sequence Learning for COVID-19 Forecasting

**Sercan Ö. Arık, Chun-Liang Li, Jinsung Yoon, Rajarishi Sinha, Arkady Epshteyn,
Long T. Le, Vikas Menon, Shashank Singh, Leyou Zhang, Martin Nikoltchev,
Yash Sonthalia, Hootan Nakhost, Elli Kanal, Tomas Pfister**
Google Cloud AI
{soarik,chunliang,jinsungyoon,sinharaj,aepshtey,longtle,vikasmenon,
shashanksi,leyouz,mnikoltchev,yashks,hootan,ekanal,tpfister}@google.com

## Abstract

We propose a novel approach that integrates machine learning into compartmental
disease modeling (e.g., SEIR) to predict the progression of COVID-19. Our model
is explainable by design as it explicitly shows how different compartments evolve
and it uses interpretable encoders to incorporate covariates and improve perfor-
mance. Explainability is valuable to ensure that the model's forecasts are credible
to epidemiologists and to instill confidence in end-users such as policy makers
and healthcare institutions. Our model can be applied at different geographic
resolutions, and we demonstrate it for states and counties in the United States. We
show that our model provides more accurate forecasts compared to the alternatives,
and that it provides qualitatively meaningful explanatory insights.

## 1 Introduction

The rapid spread of COVID-19, the disease caused by the SARS-CoV-2 virus, has had a significant
impact on humanity. Accurately forecasting the progression of COVID-19 can help (i) healthcare
institutions to ensure sufficient supply of equipment and personnel to minimize fatalities, (ii) policy
makers to consider potential outcomes of their policy decisions, (iii) manufacturers and retailers to
plan their business decisions based on predicted attenuation or recurrence of the pandemic and (iv)
the general populace to have confidence in the choices made by the above actors.

Data is one of the greatest assets of the modern era, including for healthcare [1]. We aim to exploit
this abundance of data for COVID-19 forecasting. From available healthcare supply to mobility
indices, many information sources are expected to have predictive value for forecasting the spread of
COVID-19. Data-driven time-series forecasting has enjoyed great success, particularly with advances
in deep learning [2, 3, 4]. However, several features of the current pandemic limit the success of
standard time-series forecasting methods:

- Because there is no close precedent for the COVID-19 pandemic, it is necessary to integrate
  existing data with priors based on epidemiological knowledge of disease dynamics.
- The data generating processes are non-stationary because progression of the disease influences
  public policy and individuals' public behaviors, and vice versa.
- There are many potential sources of data, but their causal impact on the disease is unclear, and
  their impact on the progression of the disease is unknown.
- The problem is non-identifiable as most infected can be undocumented.
- Data sources are noisy due to reporting issues or due to data collection problems.
- Beyond accuracy, explainability is desired – the users, either from healthcare or policy or business
  angles, should be able to interpret the results in a meaningful way for optimal strategic planning.

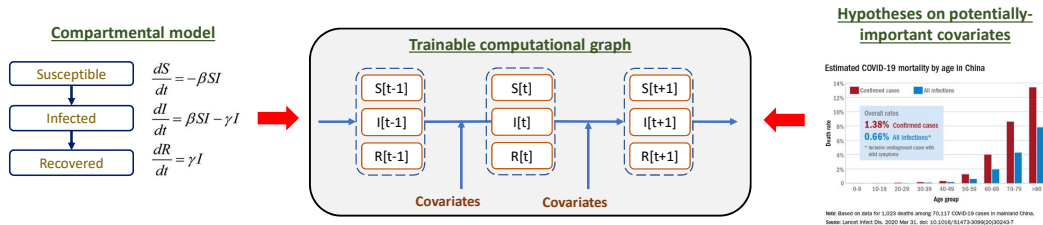

Figure 1: Our approach is based on distilling the inductive bias from compartmental models (as exemplified here for the popular SIR, Susceptible-Infected-Recovered, model) into a computational graph, where the transitions depend on the related covariates.

Compartmental models, such as the SIR and SEIR [5] models, are widely used for disease modeling by healthcare and public authorities. Such models represent the number of people in each of the compartments (see Fig. 1) and model the transitions between them via differential equations. Compartmental models often have several shortcomings: (i) only few learnable parameters resulting low model capacity (ii) non-stationary dynamics due to static rates in the differential equations; (iii) no covariates to extract information; (iv) assumptions on well-mixed compartments, i.e. each individual is statistically identical to others in the same compartment [6]; (v) no information sharing across time or geography, and (vi) non-identifiability – identical results may arise from different parametrizations [7].

While preserving interpretability for domain experts, we aim for accurate forecasts that go beyond the capabilities of standard compartmental models by utilizing rich datasets with high temporal and spatial granularity. Our approach is based on integrating covariate encoding into compartment transitions to extract relevant information via end-to-end learning (Fig. 1). In this way, we provide an inherently interpretable model that reflects the inductive biases of epidemiology. To get high accuracy, we introduce several innovative contributions:

1. We extend the standard SEIR model to also include compartments for undocumented cases and hospital resource usage. Our end-to-end modeling framework can infer meaningful estimates for undocumented cases even if there is no direct supervision for them.
2. The disease dynamics vary over time, e.g., as mobility reduces, the spreading would decay. To accurately reflect such dynamics, we propose time-varying encoding of the covariates.
3. We propose learning mechanisms to improve generalization while learning from limited training data, using (i) masked supervision from partial observations, (ii) partial teacher-forcing to minimize error propagation, (iii) regularization and (iv) cross-location information-sharing.

We demonstrate our approach for COVID-19 forecasting for the United States (US), the country that has suffered from the highest number of confirmed cases and deaths as of October 2020. For both at State- and County-level granularities, we show that our model outperforms commonly-used alternatives. Beyond accurate forecasts, we show how our model can be used for insights towards better understanding of COVID-19 pandemic.

## 2   Related work

**Compartmental models:** Using compartmental models [8] for infectious diseases can be dated back to [5], which has three compartments including susceptible, infected and recovered. Several infectious diseases, including COVID-19, manifest an incubation period during which an individual is infected, but are not yet spreaders. To this end, the Exposed (E) compartment is employed, yielding the SEIR model [9]. Beyond these basic types of compartment models, several other types of compartment models have been used, such as granular infections [10] and undocumented compartments [11]. A mixture of state-space model [12] in machine learning is presented in [13].

**Integrating covariates into compartmental models:** Policy changes such as travel bans or public restrictions have a marked, if local, effect on the disease progression. [14] designs a model that predict the effect of travel restrictions on the disease spread in China. [15] uses a modified SEIR model with mobility covariates to show the impact of interventions in the US. [16] presents a Bayesian hierarchical model for the effect of non-pharmaceutical interventions on COVID-19 in Europe. Such studies have typically been limited to the impact of one or two covariates, while our method models numerous static and time-varying ones in conjunction.

**Disease modeling using machine learning:** Apart from compartmental models, a wide variety of methods exist for modeling infectious disease. These include diffusion models [17], agent-based models [18], and cellular automata [19]. With the motivation of data-driven learning, some also integrate covariates into disease modeling, e.g. using LSTM-based models [20, 21, 22].

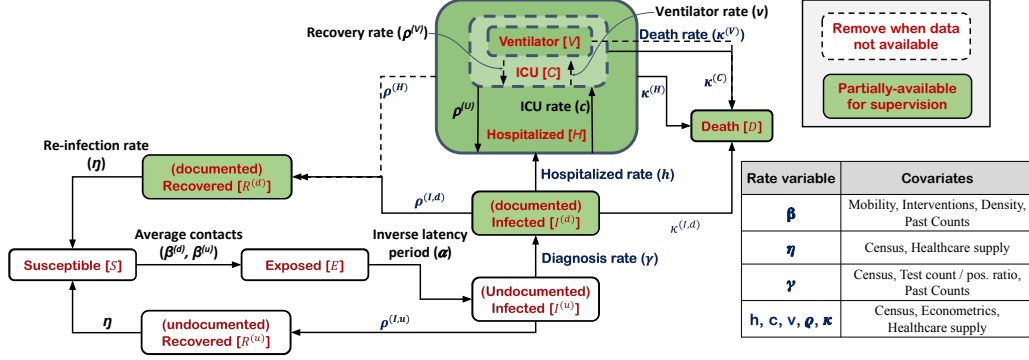

Figure 2: The modeled compartments and the corresponding covariates, with the legend on the right.

**Learning from data and equations:** Strong inductive biases can improve machine learning. One type of such bias is the set of equations between input and output, particularly common in physics or chemistry. To incorporate the inductive bias of equations, several recent works [23, 24, 25, 26] have studied parametric approaches, as in our paper, where trainable models are incorporated to model only certain terms in the equations, while the equations still govern the end-to-end relationships.

**Other COVID-19 forecasting works:** Different works have adopted compartmental models for COVID-19 forecasting via modeling different comparments, such as YYG [27]. However, they do not leverage additional covariates. IHME [28] is based on fitting a curve to model the non-linear mixing effects, which does not explicitly model the transitions between the compartments. LANL [29] is based on statistical-dynamical growth modeling for the susceptible and infected cases.

## 3  Proposed compartmental model for COVID-19

We adapt the standard SEIR model with some major changes, as shown in Fig. 2:

- **Undocumented infected and recovered compartments**: Recent studies suggest that majority of the infected people are not detected and they dominate disease spreading[1] [30, 31] (as the documented ones are either self-isolated or hospitalized) An undocumented infected individual is able to spread the disease, until being documented or recovered without being undocumented.
- **Hospitalized, ICU and ventilator compartments:** We introduce compartments for the people who are hospitalized, in the ICU, or on a ventilator, as there is a demand to model these [32] and there is partially-available observed data to be used for supervision.
- **Partial immunity**: To date, there is no scientific consensus on what fraction of recovered cases demonstrate immunity to future infection. Due to reports of reinfection [33] we model the rate of reinfection from recovered compartments (though our model infers low reinfection rates).
- **Other Assumptions**: We assume the published COVID-19 death counts are coming from documented cases, not undocumented. Also, we assume that the entire population is invariant, i.e. births and non-Covid deaths are negligible in comparison to the entire population. Last, by data publishing frequency, we assume a fixed sampling interval of 1 day.

Table 1: Modeled compartments.

| Compartment | Description | Compartment | Description |
|---|---|---|---|
| $S$ | Susceptible | $R^{(u)}$ | Recovered undocumented |
| $E$ | Exposed | $H$ | Hospitalized |
| $I^{(d)}$ | Infected documented | $C$ | In intensive care unit (ICU) |
| $I^{(u)}$ | Infected undocumented | $V$ | On ventilator |
| $R^{(d)}$ | Recovered documented | $D$ | Death |

The modeled compartments are shown in Table 1. For a compartment $X$, $X_i[t]$ denotes the number of individuals in that compartment at location $i$ and time $t$. $N[t]$ denotes the total population. Fig. 2 describes transition rate variables used to relate the compartments, via the equations (we omit the index $i$ for concision):

$$
\begin{aligned}
S[t] - S[t-1] &= -(\beta^{(d)}I^{(d)}[t-1] + \beta^{(u)}I^{(u)}[t-1])\tfrac{S[t-1]}{N[t-1]} + \eta(R^{(d)}[t-1] + R^{(u)}[t-1]), \\
E[t] - E[t-1] &= (\beta^{(d)}I^{(d)}[t-1] + \beta^{(u)}I^{(u)}[t-1])\tfrac{S[t-1]}{N[t-1]} - \alpha E[t-1], \\
I^{(u)}[t] - I^{(u)}[t-1] &= \alpha E[t-1] - (\rho^{(I,u)} + \gamma)I^{(u)}[t-1], \\
I^{(d)}[t] - I^{(d)}[t-1] &= \gamma I^{(u)}[t-1] - (\rho^{(I,d)} + \kappa^{(I,d)} + h)I^{(d)}[t-1], \\
R^{(u)}[t] - R^{(u)}[t-1] &= \rho^{(I,u)}I^{(u)}[t-1] - \eta R^{(u)}[t-1], \\
R^{(d)}[t] - R^{(d)}[t-1] &= \rho^{(I,d)}I^{(d)}[t-1] + \rho^{(H)}(H[t-1] - C[t-1]) - \eta R^{(d)}[t-1], \\
H[t] - H[t-1] &= hI^{(d)}[t-1] - (\kappa^{(H)} + \rho^{(H)})(H[t-1] - C[t-1]) - \kappa^{(C)}(C[t-1] - V[t-1]) - \kappa^{(V)}V[t-1], \\
C[t] - C[t-1] &= c(H[t-1] - C[t-1]) - (\kappa^{(C)} + \rho^{(C)} + v)(C[t-1] - V[t-1]) - \kappa^{(V)}V[t-1], \\
V[t] - V[t-1] &= v(C[t-1] - V[t-1]) - (\kappa^{(V)} + \rho^{(V)})V[t-1], \\
D[t] - D[t-1] &= \kappa^{(V)}V[t-1] + \kappa^{(C)}(C[t-1] - V[t-1]) + \kappa^{(H)}(H[t-1] - C[t-1]) + \kappa^{(I,d)}I^{(d)}[t-1],
\end{aligned}
$$

**Corollary: Basic reproduction number** An analysis of our compartmental model using the Next-Generation Matrix method [34] yields the effective reproductive number (spectral radius) as:

$$
R_e = \frac{\beta^{(d)}\gamma + \beta^{(u)}(\rho^{(I,d)} + \kappa^{(I,d)} + h)}{(\gamma + \rho^{(I,u)}) \cdot (\rho^{(I,d)} + \kappa^{(I,d)} + h)}. \tag{1}
$$

Please see Appendix for derivations. Note that when $\gamma = 0$, our compartmental model reduces to the standard SEIR model with the undocumented infected and recovered. In this case, $R_0 = \beta^{(u)}/\rho^{(I,u)}$.

## 4 Encoding covariates

**Time-varying modeling of variables:** Instead of using static rate variables across time to model compartment transitions as in standard compartmental models, there should be time-varying functions that map them from known observations. For example, if human mobility decreases over time, the $S \to E$ transition should reflect that. Consequently, we propose replacing all static rate variables with learnable functions that output their value from the related static and time-varying covariates at each location and timestep. We list all the covariates used for each rate variable in the Appendix. We note that learnable encoding of variables still preserves the inductive bias of the compartmental modeling framework while increasing the model capacity via learnable encoders.

**Interpretable encoder architecture:** In addition to making accurate forecasts, it is valuable to understand how each covariate affects the model. Such explanations greatly help users from healthcare and public sector to understand the disease dynamics better, and also help model developers to ensure the model is learning appropriate dynamics via sanity checks with known scientific studies or common knowledge. To this end we adopt a generalized additive model [35] for each variable $v_i$ from Table 2 based on additional *covariates* $\text{cov}(v_i, t)$ at different time $t$. The covariates we consider include (i) the set of static covariates $\mathcal{S}$, such as population density, and (ii) $\{f[t-j]\}_{f \in \mathcal{F}_i, j=1,\dots,k}$ the set of time-varying covariates (features) $\mathcal{F}_i$ with the observation from $t-1$ to $t-k$, such as mobility. Omitting individual feature interactions and applying additive aggregation, we obtain

$$
v_i[t] = v_{i,L} + (v_{i,U} - v_{i,L}) \cdot \sigma\left(c + b_i + \mathbf{w}^\top \text{cov}(v_i, t)\right), \tag{2}
$$

where $v_{i,L}$ and $v_{i,U}$ are the lower and upper bounds of $v_i$ for all $t$, $c$ is the global bias, $b_i$ is the location-dependent bias. $\mathbf{w}$ is the trainable parameter, and $\sigma()$ is the sigmoid function to limit the range to $[v_{i,L}, v_{i,U}]^2$, which is important to stabilize training and avoid overfitting. We note that although Eq. (2) denotes a linear decomposition for $v_i[t]$ at each timestep, the overall behavior is still highly non-linear due to the relationships between compartments.

**Covariate forecasting:** The challenge of using Eq. (2) for future forecasting is that some time-varying covariates are not available for the entire forecasting horizon. Assume we have the observations of covariates and compartments until $T$, and we want to forecast from $T+1$ to $T+\tau$. To forecast $v_i[T+\tau]$, we need the time varying covariates $f[T+\tau-k : T+\tau-1]$ for $f \in \mathcal{F}_i$, but some of them are not observed when $\tau > k$. To solve this issue, we propose to forecast $f[T+\tau-k : T+\tau-1]$ based on their own past observations until $T$, which is a standard one dimensional time series forecasting for a given covariate $f$ at a given location. In this paper, we use a standard XGBoost model [36] which inputs time-series features.[3]

**Information-sharing across locations:** Some aspects of the disease dynamics are location-dependent while others are not. In addition, data availability varies across locations – there may be limited observations to learn the impact of a covariate. A model able to learn both location dependent

and independent dynamics is desirable. Our encoders in Eq. (2) partially capture location-shared dynamics via shared $\mathbf{w}$ and the global bias $c$. To allow the model capture remaining location-dependent dynamics, we introduce the local bias $b_i$. A challenge is that the model could ignore the covariates by encoding all information into $b_i$ during training. This could hurt generalization as there would not be any information-sharing on how static covariates affect the outputs across locations. Thus, we introduce a regularization term $L_{ls} = \lambda_{ls} \sum_i |b_i|^2$ to encourage the model to leverage covariates and $c$ for information-sharing instead of relying on $b_i$. Without $L_{ls}$, we observe that the model would use the local bias more than the encoded covariates, and suffers from poorer generalization.

## 5 End-to-end training

---

**Algorithm 1** Pseudo-code for training the proposed model

---

**Inputs:** Forecasting horizon $\tau$, compartment observations $Q_i, H_i, C_i, V_i, D_i, R_i$ from $T_s$ until $T$, the number of fine tuning iterations $F$, loss coefficients $\lambda_{R_e}$ and $\lambda_{ls}$.

Initialize trainable parameters $\theta = \{\mathbf{w_i}, c, b_i\}$, and initial conditions for the compartments $\hat{E}[0]$, $\hat{I}^{(d)}[0], \hat{I}^{(u)}[0], \hat{R}^{(d)}[0], \hat{R}^{(u)}[0], \hat{H}[0], \hat{C}[0], \hat{V}[0], \hat{D}[0]$

Split $\tau$ day validation $Y_i[T - \tau : T]$ for all locations $i$, where $Y \in \{Q, H, C, V, D, R^{(d)}\}$

**while** until convergence **do**

    Sample initial conditions $E_i[0], I_i^{(d)}[0], I_i^{(u)}[0], R_i^{(d)}[0], R_i^{(u)}[0], H_i[0], C_i[0], V_i[0], D_i[0]$

    $\theta \leftarrow \theta - \text{RMSProp}(\nabla_\theta \mathcal{L}(T_s, T - \tau - 1))$

    Update the optimal parameters: $\theta_{opt} = \theta$ if $L_{fit}[T - \tau : T]$ is the current-best

**Final fine-tuning:** fine-tune with joint training and validation data:

$\theta \leftarrow \theta_{opt}$

**for** $F$ iterations **do**

    $\theta \leftarrow \theta - \text{RMSProp}(\nabla_\theta \mathcal{L}(T_s, T))$

    Update the optimal parameters: $\theta_{opt} = \theta$ if $L_{fit}[T - \tau : T]$ is the currently best

**Output:** Return $\theta_{opt}$

---

**Learning from partially-available observations:** Fitting would have been easy with observations for all compartments, however, we only have access to some. For instance, $I^{(d)}$ is not given in the ground truth of US data but we instead have, $Q$, the total number of confirmed cases, that we use to supervise $I^{(d)} + R^{(d)} + H + D$. Note that $R^{(ud)}, I^{(ud)}, S, E$ are not given as well. Formally, we assume availability of the observations $Y[T_s : T]$[4], for $Y \in \{Q, H, C, V, D, R^{(d)}\}$, and consider forecasting the next $\tau$ days, $\hat{Y}[T + 1 : T + \tau]$.

**Fitting objective:** There is no direct supervision for training encoders, while they should be learned in an end-to-end way via the aforementioned partially-available observations. We propose the following objective for range $[T_s, T_e]$:

$$L_{fit}[T_s : T] = \sum_{Y \in \{Q,H,C,V,D,R^{(d)}\}} \lambda_Y \sum_{t=T_s}^{T-\tau} \sum_{i=1}^{\tau} \frac{\mathbb{I}(Y[t+i])}{\sum_j \mathbb{I}(Y[j]) \cdot Y[j]} \cdot q(t+i-T_s; z) \cdot L(Y[t+i], \hat{Y}[t+i]).$$

$$(3)$$

$\mathbb{I}(\cdot) \in \{0, 1\}$ indicates the availability of the $Y$ to allow the training to focus only on available observations. $L(,)$ is the loss between the ground truth and the predicted values (e.g., $\ell_2$ or quantile loss), and $\lambda_Y$ are the importance weights to balance compartments due to its robustness (e.g., $D$ is much more robust than others). Lastly, $q(t; z) = \exp(t \cdot z)$ is a time-weighting function (when $z = 0$, there is no time weighting) to favor more recent observations with $z$ as a hyperparameter.

**Constraints and regularization:** Given the limited dataset size, overfitting is a concern for high-capacity encoders trained on insufficient data. In addition to limiting the model capacity with the epidemiological inductive bias, we further apply regularization to improve generalization to unseen future data. An effective regularization is constraining the effective reproduction number $R_e$ as derived in Eq. (1). There are rich literature in epidemiology on $R_e$ to give us good priors on the range of the number should be. For a reproduction number $R_e[t]$ at time $t$, we consider the regularization

$$L_{R_e}[T_s : T] = \sum_{t=T_s}^{T} \exp\left((R_e[t] - R)_+\right),$$

where $R$ is a prespecified *soft* upper bound. The regularization favors the model with $R_e$ in a reasonable range in addition to good absolute forecasting numbers. In the experiment, we set $R = 5$ without further tuning. Last, ignoring the perturbation of a small local window, the trend of forecast should usually be smooth. One commonly used smoothness constraint, is on the first-order difference. We call it as *velocity*, which is defined as $v_Y[t] = (Y[t] - Y[t - k])/k$. The first-order constraint encourage $v_Y[t] \approx v_Y[t - 1]$, which causes linear forecasting, and cannot capture the rapid growing cases. Instead, we relax the smoothness to be on the second order difference. We called it as *acceleration*, which is defined as $a_Y[t] = v_Y[t] - v_Y[t - 1]$. The regularization is

$$L_{acc}[T_s : T] = \sum_{Y \in \{Q, D\}} \sum_{t=T_s+1}^{T} (a_Y[t] - a_Y[t - 1])^2$$

The final objective function is

$$\mathcal{L}(T_s, T) = L_{fit}[T_s : T] + \lambda_{ls} \cdot L_{ls} + \lambda_{R_e} \cdot L_{R_e}[T_s : T] + \lambda_{acc} \cdot L_{acc}[T_s : T], \qquad (4)$$

where $L_{ls} = \lambda_{ls} \sum_i |b_i|^2$ as discussed in Sec. 4.

**Partial teacher forcing:** The compartmental model presented in Sec. 3 produces the future propagated values from the current timestep. During training, we have access to the observed values for $Y \in \{Q, H, C, V, D, R^{(d)}\}$ at every timestep, which we could condition the propagated values on, commonly-known as teacher forcing [37] to mitigate error propagation. At inference time, however, ground truth beyond the current timestep $t$ is unavailable, hence the predictions should be conditioned on the future estimates. Using solely ground-truth to condition propagation would create a train-test mismatch. In the same vein of past research to mix the ground truth and predicted data to condition the projections on [38], we propose partial teacher forcing, simply conditioning $(1 - \nu\mathbb{I}\{Y[t]\})Y[t] + \nu\mathbb{I}\{Y[t]\})\hat{Y}[t]$, where $\mathbb{I}\{Y[t]\} \in \{0, 1\}$ indicates whether the ground truth $Y[t]$ exists and $\nu \in [0, 1]$. In the first stage of training, we use teacher forcing with $\nu \in [0, 1]$, which is a hyperparameter. For fine-tuning (please see below), we use $\nu = 1$ to unroll the last $\tau$ steps to mimic the real forecasting scenario.

**Model fitting and selection:** The training pseudo code is presented in Algorithm 1. We split the observed data into training and validation with the last $\tau$ timesteps to mimic the testing scenario. We use the training data for optimization of the trainable degrees of freedom, collectively represented as $\theta$, while the validation data is used for early stopping and model selection. Once the model is selected, we fix the hyperparameters and run fine-tuning on joint training and validation data, to not waste valuable recent information by using it only for model selection. For optimization, we use RMSProp as it is empirically observed to yield lower losses compared to other algorithms and providing the best generalization performance.

## 6    Experiments

**Ground truth data:** We conduct all experiments on US COVID-19 data. The primary ground truth data for the progression of the disease, for $Q$ and $D$, are from [39] as used by several others, e.g. [28]. They obtain the raw data from the state and county health departments. Because of the rapid progression of the pandemic, past data has often been restated, or the data collection protocols have been changed. Ground truth data for the $H$, $C$ and $V$ (see Fig. 1) are obtained from [40]. Note that there are significant restatements of the past observed counts in the data, so we use the reported numbers on the prediction date for training (although later we know the restated past ground truth), and the reported numbers $\tau$ days after prediction date for evaluation, to be completely consistent with other models for fair comparison.

**Covariates:** The progression of COVID-19 is influenced by a multitude of factors, including relevant properties of the population, health, environmental, hospital resources, demographics and econometrics indicators. Time-varying factors such as population mobility, hospital resource usage and public policy decisions can also be important. However, indiscriminately incorporating a data source may have deleterious effects. Thus, we curate our data sources to limit them to one source in each category of factors that may have predictive power at the corresponding transition. We use datasets from public sources (please see Appendix for details). We apply forward- and backward-filling imputation (respectively) for time-varying covariates, and median imputation for static covariates. Then, all covariates are normalized to be in [0, 1], considering statistics across all locations and time-steps.

**Training:** We implement Algorithm 1 in TensorFlow at state- and county-levels, using $\ell_2$ loss for point forecasts. We employ [41] for hyperparameter tuning (including all the loss coefficients,

learning rate, and initial conditions) with the objective of optimizing for the best validation loss, with 400 trials and we use $F = 300$ fine-tuning iterations. We choose the compartment weights $\lambda^D = \lambda^Q = 0.1$, $\lambda^H = 0.01$ and $\lambda^{R^{(d)}} = \lambda^C = \lambda^V = 0.001$.[5] At county granularity, we do not have published data for $C$ and $V$, so, we remove them along with their connected variables.

## 6.1 Results

Table 2: $\tau$-day average MAE for forecasting the number of deaths at state-level. Since benchmark models from covid19-forecast-hub repository release forecasts at different dates and horizons, not all models have predictions for all prediction dates/horizons (indicated by "—"). **Bold** indicates the best.

| Pred. horizon $\tau$ (days) | Pred. date | Ours | CU | LANL | UT | YYG |
|---|---|---|---|---|---|---|
| 14 | 05/19/2020 | **35.8** | 71.4 | 45.3 | 43.7 | 46.5 |
| | 05/26/2020 | **29.4** | 58.5 | 36.3 | 43.8 | 37.7 |
| | 06/02/2020 | 32.8 | 86.1 | 33.5 | 35.1 | **26.5** |
| | 06/09/2020 | 28.8 | 71.0 | 34.7 | 33.5 | **22.3** |
| | 06/16/2020 | **31.4** | 79.6 | 50.8 | 48.9 | 32.1 |
| | 06/23/2020 | **63.8** | 134.7 | 85.8 | 67.7 | 64.2 |
| | 06/30/2020 | 46.5 | 152.1 | 48.6 | **34.1** | 35.1 |

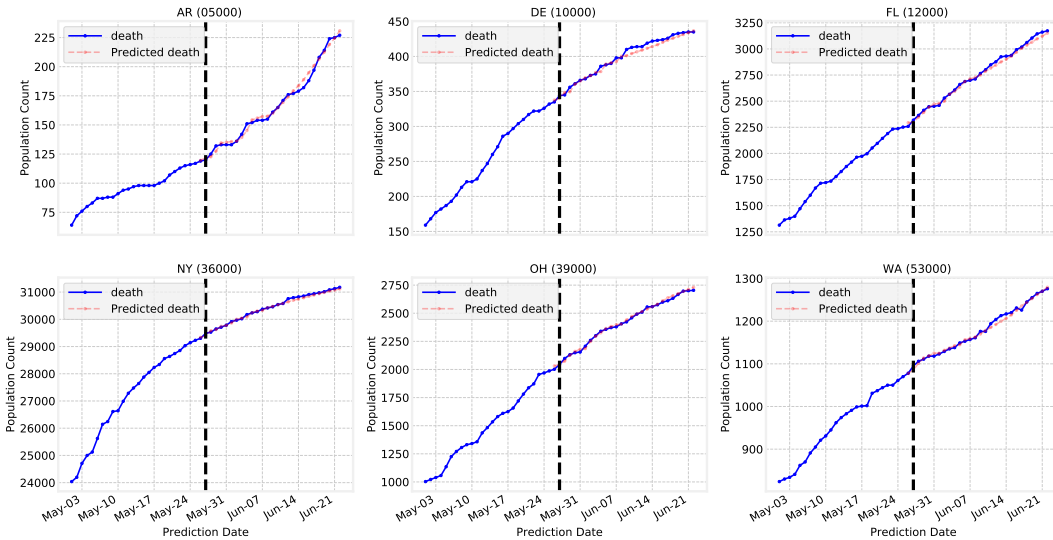

Figure 3: Ground-truth vs. predicted 14-day death forecasts for 6 states: AR, DE, FL, NY, OH, and WA on 06/09/2020.

**State-level forecasts:** Fig. 3 exemplifies the forecasting performance of our model on 4 states. We compare our method to widely-used benchmarks for state-level prediction of the number of deaths in each US state. Specifically, we report comparisons with Columbia University (CU) model [42], the GrowthRate model from Los Alamos National Laboratory (LANL) [29], UT-Austin (UT) model [43] and the YYG model [27]. CU is a metapopulation SEIR model with a selection mechanism among the different generated scenarios for interventions. LANL is based on statistical-dynamical growth modeling for the underlying numbers of susceptible and infected cases. UT makes predictions assuming that social distancing patterns, as measured by anonymized mobile-phone GPS traces, using a Bayesian multilevel negative binomial regression model. YYG is an SEIR model with learnable parameters and accounts for reopenings. The parameters are fit using hyperparameter optimization. Unlike ours, YYG uses fixed (time-invariant) rates as SEIR parameters and is limited to modeling standard SEIR compartments. Note that, in contrast to usual benchmarks, these models may change significantly between forecast dates. Table 2 shows comparisons for different prediction dates and forecasting horizons $\tau$. Our model is consistently accurate across phases of the pandemic and outperforms all others except YYG by a large margin. YYG is merely optimized for the number of deaths, whereas our model jointly predicts all the compartments while being explainable. Fig. 3 exemplifies our forecasting on different states, and shows our model can forecast well on different scale of reported deaths.

Table 3: $\tau$-day average MAE for 14-day forecasting of the number of deaths for county-level forecasts.

| Pred. horizon $\tau$ (days) | Pred. date | Ours | Berkeley CLEP |
|---|---|---|---|
| 14 | 05/19/2020 | **1.19** | 1.91 |
| | 06/09/2020 | **1.02** | 1.79 |
| | 05/19/2020 | **1.80** | 3.24 |
| | 05/26/2020 | **1.56** | 3.10 |
| | 06/09/2020 | **1.36** | 3.20 |
| | 06/16/2020 | **1.37** | 3.32 |

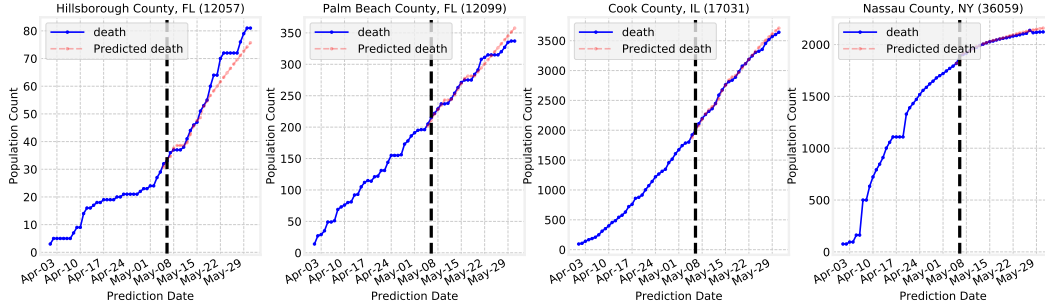

Figure 4: Ground-truth vs. predicted 14-day death forecasts for 4 counties: Hillsborough Co., FL; Palm Beach Co., FL; Cook Co., IL; Nassau Co., NY on 05/19/2020.

**County-level forecasts:** Table 3 shows the performance of our model on all (more than 3000) US counties. Compared with state-level forecasting, it is more challenging due to sparse observations. We compare our method to predictions by Berkeley Yu model [44] for the number of deaths in each US county. The model comprises several predictors (including exponential and linear) and ensemble their forecasts resulting Combined Linear and Exponential Predictors (CLEP). The setting to compare the county is same as state-level comparisons. Table 3 demonstrates that our model yields much lower error compared to the Berkeley CLEP model. Fig. 4 exemplifies the prediction for a few counties.

## 6.2 Ablation studies

Table 4: $\tau$-day average MAE for 14-day forecasting of the number of deaths at state-level.

| Models / Prediction date | 05/25/2020 | 06/01/2020 | 06/08/2020 |
|---|---|---|---|
| Standard SEIR compartments (*w/o* encoder) | 87.3 | 76.1 | 71.0 |
| Standard SEIR compartments (*with* encoder) | 50.0 | 37.5 | 39.3 |
| Our model (*w/o* encoder) | 69.2 | 36.8 | 28.7 |
| Our model *w/o* fine-tuning | 94.1 | 78.7 | 65.8 |
| Our model *w/o* partial teacher forcing | 792.6 | 1903.8 | 1289.7 |
| **Our model** | **32.9** | **23.8** | **26.5** |

Table 4 presents the major results for ablation cases. For these ablation studies, to merely focus on the impact of the model changes, we use the most recently-updated data for both training and evaluation. We observe the significant benefits of (i) learning rates from covariates with encoders, (ii) modeling extra compartments and supervision from $H$, $C$ and $V$, (iii) partial teacher forcing and (iv) final-fine tuning, adapting to the most recent data after model selection based on validation.

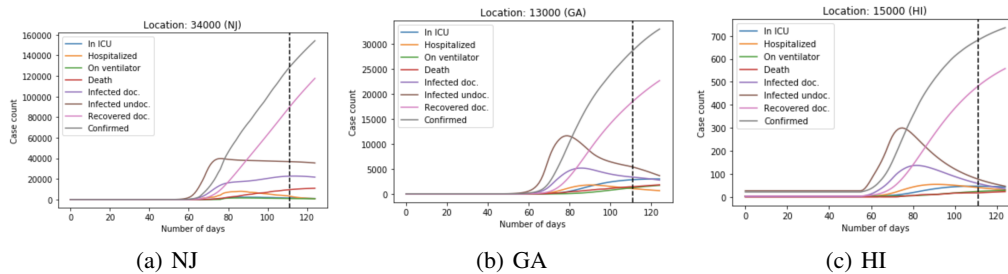

(a) NJ      (b) GA      (c) HI

Figure 5: Fitted compartments for (a) NJ, (b) GA and (c) HI, where vertical lines show the forecasting starting timestep. Note that infected values are not cumulative, thus decay over time while the confirmed keeps increasing. These can be used to gain insights in disease evolution, e.g. we observe the increasing trend of the number of confirmed cases more sharply in NJ, whereas it is saturating in HI, due to the sharp decrease in the number of infected people after the peak.

## 6.3 Extracting explainable insights

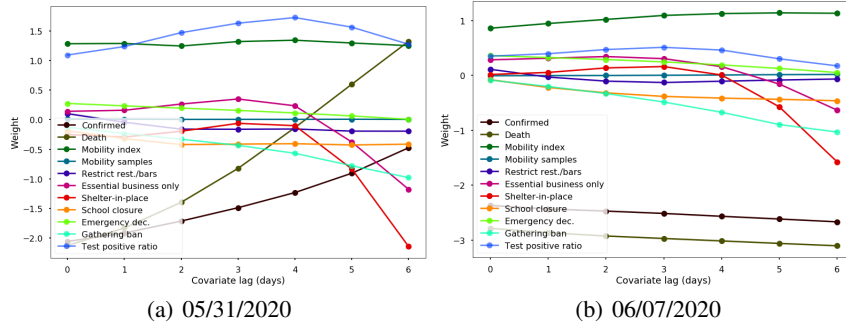

(a) 05/31/2020           (b) 06/07/2020

Figure 6: Learned weights of covariates for $\beta^{(u)}$, for 7-day state-level forecasting models on 05/31/2020 and 06/07/2020. Mobility index consistently has highly-positive impact on $\beta^{(u)}$, while gathering bans, school closures and shelter-in-place interventions have highly-negative impact. The magnitude of the weights for interventions get larger after a lag of few days.

The interpretability of our model is two fold. First, we model the compartments explicitly, thus our model provides insights into how the disease evolves. Fig 5 shows the fitted curves that can be used to infer important insights on where the peaking occurs, or the current decay trends. We observe the ratio of undocumented to documented infected at different phases, as well as the amount of increase/decrease for each compartment. Second, our model uses interpretable encoders, as discussed in Sec. 4. Ignoring co-linearity between covariates, rough insights can be inferred. Fig. 6 shows the learned weights of the time-varying covariates for $\beta^{(u)}$. The weights of the past days seem similar – the model averages them with slight decay in trends. For intervention covariates, the largest weights occur after a lag of a few days, suggesting their effectiveness after some lag. The positive weights of the mobility index, and negative weights of public interventions are clearly observed. Similar analysis can be performed on other variables as well. For $\gamma$, we observe the positive correlation of the positive ratio of tests. For static covariates, the insights are less apparent, but we observe meaningful learned patterns like the positive correlation of the number of households on public assistance or food stamps, population density and 60+ year old population ratio, on death rates.

## 6.4 Forecasting prediction intervals

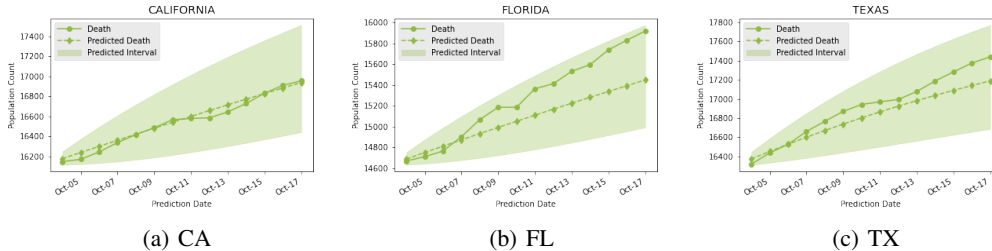

(a) CA           (b) FL           (c) TX

Figure 7: Prediction intervals for 14-day forecasting. We use the 10-th and the 90-th quantile prediction as the the lower and the upper bound of prediction intervals, respectively.

Besides point forecasts, prediction intervals could be helpful for healthcare and public policy planners, to consider a range of possible scenarios. Our framework allows the capability of modeling prediction interval forecasts, for which we replacing the L2 loss with weighted interval loss [45] in Eq. (3) and mapping the scalar propagated values to the vector of quantile estimates. For this mapping, we use the features $Y[t]/\hat{Y}[t]$ and $\mathbb{I}\{Y[t]\}$ for $T - \tau \leq t \leq T - 1$. We obtain the quantiles applying a linear kernel on these features, followed by ReLU and cumulative summation (to guarantee monotonicity of quantiles) and lastly normalization (to match the median to the input scalar point forecast from the proposed model). Fig. 7 exemplifies well-calibrated prediction interval forecasts – the ranges tend to be wider when there are non-smooth behaviors in data.

## 7 Conclusions

We propose an approach to modelling infectious disease progression by incorporating covariates into a domain-specific encoding, understandable by experts. We compare predictions for this novel model with state-of-the-art models and show that disaggregating the infected compartment into sub-compartments relevant to decision-making can make the model more useful to decision-makers.

# 8 Acknowledgements

Contributions of Nathanael C. Yoder, Michael W. Dusenberry, Dario Sava, Jasmin Repenning, Andrew Moore, Matthew Siegler, Ola Rozenfeld, Isaac Jones, Rand Xie, Brian Kang, Vishal Karande, Shane Glass, Afraz Mohammad, David Parish, Ron Bodkin, Hanchao Liu, Yong Li, Karthik Ramasamy, Priya Rangaswamy, Andrew Max, Tin-yun Ho, Sandhya Patil, Rif A. Saurous, Matt Hoffman, Peter Battaglia, Oriol Vinyals, Jeremy Kubica, Jacqueline Shreibati, Michael Howell, Meg Mitchell, George Teoderci, Kevin Murphy, Helen Wang, Tulsee Doshi, Garth Graham, Karen DeSalvo, and David Feinberg are gratefully acknowledged.

## Broader Impact

COVID-19 is an epidemic that is affecting almost all countries in the world at the moment. As of the first week of June, more than 6.5 million people have been infected, resulting in more than 380k fatalities unfortunately. The economical and sociological impacts of COVID-19 are significant, and will be felt for many years to come.

Forecasting of the severity of COVID-19 is crucial, for healthcare providers to deliver the healthcare support for those who will be in the most need, for governments to take the most optimal policy actions while minimizing the negative impact of the outbreak, and for business owners to make crucial decisions on when and how to restart their businesses. With the motivation of helping all these actors, we propose a machine learning-based forecasting model that significantly outperforms any alternative methods, including the ones used by the healthcare providers and public sector. Not only are our forecasts far more accurate, our model is explainable by design. It is aligned with how epidemiology experts approach the problem, and the machine learnable components shed light on what data features have the most impact on the outcomes. These would be crucial for data-driven understanding of COVID-19, that can help domain experts for effective medical and public health decision-making.

Besides COVID-19 forecasting, our approach is in the direction of ingesting data-driven learning while using the inductive bias of differential equations, while representing the input-output relationships at a system-level. Not only infectious disease modeling, but numerous scientific fields that use such equations, such as Physics, Environmental Sciences, Chemistry etc. are expected to benefit from our contributions.

## Footnotes

[1][11] estimates that $> 80\%$ of cases in China were undocumented during the early phase of the pandemic.

[2]We use $v_{i,L}{=}0$ for all variables, $v_{i,U} = 1$ for $\beta$, 0.2 for $\alpha$, 0.001 for $\eta$ and 0.1 for others.

[3]We used the lagged features of the past 7 days plus the 2 weeks ago, and mean/max in the windows of sizes of 3, 5, 7, 14 and 21 days.

[4]We use the notation $S_i[T_s : T]$ to denote all timesteps between $T_s$ (inclusive) and $T$ (inclusive).

[5]Our results are not highly sensitive to these.

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
