[Supplementary Material]

# Supplementary Materials: Interpretable Sequence Learning for COVID-19 Forecasting

**Sercan Ö. Arık, Chun-Liang Li, Jinsung Yoon, Rajarishi Sinha, Arkady Epshteyn,**
**Long T. Le, Vikas Menon, Shashank Singh, Leyou Zhang, Martin Nikoltchev,**
**Yash Sonthalia, Hootan Nakhost, Elli Kanal, Tomas Pfister**
Google Cloud AI
{soarik,chunliang,jinsungyoon,sinharaj,aepshtey,longtle,vikasmenon,
shashanksi,leyouz,mnikoltchev,yashks,hootan,ekanal,tpfister}@google.com

## A Datasets

As discussed, vast numbers of candidate datasets exist that could be related to the problem of COVID-19 forecasting. However, these datasets cannot be used indiscriminately. We select data sources based on whether they could have a predictive signal for the disease outcomes. Selecting multiple datasets from the same class of causes can obfuscate their predictive power. Therefore, we select datasets, one each from the classes of econometrics, demographics, mobility, non-pharmaceutical interventions, hospital resource availability, historical air quality. From each of these datasets, we further select covariates that could have an impact on the model compartments. We allow covariates to influence only those compartments (and hence transition rates) on which we posit that there exists a causal relationship (Table 1).

**Ground Truth**. We obtain primary ground truth for this work from the Johns Hopkins COVID-19 dataset [1]. Additional ground truth data that is used in the models for US states are obtained from the Covid Tracking Project [2].

**Mobility**. We posit that human mobility with a region, for work and personal reasons, has an effect on the average contact rates [3]. We use temporal mobility indices provided by Descartes labs at both state- and county-level resolutions [4]. These temporal indices are encoded to affect the average contact rates ($\beta^{(d)}$, $\beta^{(u)}$), at both the state- and county-level of geographic resolution.

**Non-Pharmaceutical Interventions**. We posit that public policy decisions restricting certain classes of population movement or interaction can have a beneficial effect on restricting the progression of the disease [5], at the state-level of geographic resolution. The interventions are presented in 6 binary valued time series indicating when an intervention has been activated in one of six categories–school closures, restrictions on bars and restaurants, movement restrictions, mass gathering restrictions, essential businesses declaration, and emergency declaration [6]. This temporal covariate is encoded into the average contact rates ($\beta^{(d)}$, $\beta^{(u)}$).

**Demographics**. We posit that the age of the individual has a significant outcome on the severity of the disease and the mortality. The Kaiser Family Foundation (On BigQuery at c19hcc-info-ext-data:c19hcc_info_public.Kaiser_Health_demographics_by_Counties_States) reports the number of individuals over the age of 60 in different US counties. We encode the effect of this static covariate into the average contact rate ($\beta^{(d)}$, $\beta^{(u)}$), the diagnosis ($\gamma$), re-infected ($\eta$), recovery ($\rho^{(I,d)}$, $\rho^{(I,u)}$, $\rho^{(H)}$, $\rho^{(C)}$, $\rho^{(V)}$) and death rates ($\kappa^{(I,d)}$, $\kappa^{H}$, $\kappa^{C}$, $\kappa^{V}$), at both the state- and county-level of geographic resolution.

**Historical Air Quality**. We posit that the historical ambient air quality in a region can have a deleterious effect on COVID-19 morbidity and mortality [7]. We use the BigQuery public dataset that comes from the US Environmental Protection Agency (EPA) that documents historical air quality indices at the county level (bigquery-public-data:epa_historical_air_quality.pm10_daily_summary).

Table 1: Covariates selected for model.

| Covariate | Variables that the covariate affect |
|---|---|
| Per capita income | $\beta^{(d)}, \beta^{(u)}, \eta, \gamma, \rho^{(I,d)}, \rho^{(I,u)}, \rho^{(H)}, \rho^{(C)}, \rho^{(V)}, h, c, v, \kappa^{(I,d)}, \kappa^H, \kappa^C, \kappa^V$ |
| Population density | $\beta^{(d)}, \beta^{(u)}, \eta, \gamma, \rho^{(I,d)}$ |
| Households on food stamps | $\eta, \rho^{(I,d)}, \rho^{(I,u)}, \rho^{(H)}, \rho^{(C)}, \rho^{(V)}, h, c, v, \kappa^{(I,d)}, \kappa^H, \kappa^C, \kappa^V$ |
| Population | All |
| Number of households | $\beta^{(d)}, \beta^{(u)}, \eta, \gamma, \rho^{(I,d)}, \rho^{(I,u)}, \rho^{(H)}, \rho^{(C)}, \rho^{(V)}, h, c, v, \kappa^{(I,d)}, \kappa^H, \kappa^C, \kappa^V$ |
| Population ratio above age 60 | $\beta^{(d)}, \beta^{(u)}, \eta, \gamma, \rho^{(I,d)}, \rho^{(I,u)}, \rho^{(H)}, \rho^{(C)}, \rho^{(V)}, h, c, v, \kappa^{(I,d)}, \kappa^H, \kappa^C, \kappa^V$ |
| Hospital rating scale | $\eta, \gamma, \rho^{(I,d)}, \rho^{(I,u)}, \rho^{(H)}, \rho^{(C)}, \rho^{(V)}, h, c, v, \kappa^{(I,d)}, \kappa^H, \kappa^C, \kappa^V$ |
| Available types of hospitals | $\eta, \rho^{(I,d)}, \rho^{(I,u)}, \rho^{(H)}, \rho^{(C)}, \rho^{(V)}, h, c, v, \kappa^{(I,d)}, \kappa^H, \kappa^C, \kappa^V$ |
| Hospital patient experience rating | $\eta, \rho^{(I,d)}, \rho^{(I,u)}, \rho^{(H)}, \rho^{(C)}, \rho^{(V)}, h, c, v, \kappa^{(I,d)}, \kappa^H, \kappa^C, \kappa^V$ |
| Air quality measures | $\beta^{(d)}, \beta^{(u)}, \eta, \kappa^{(I,d)}$; also for state model: $h, c, v, \kappa^H, \kappa^C, \kappa^V$ and for county model: $\gamma, \rho^{(I,d)}, \rho^{(I,u)}$ |
| Mobility indices | $\beta^{(d)}, \beta^{(u)}$ |
| Non-pharmaceutical interventions (state model) | $\beta^{(d)}, \beta^{(u)}$ |
| Total tests (state model) | $\gamma, h$ |
| Confirmed per Total tests | $\beta^{(d)}, \beta^{(u)}, \gamma, h$ |
| Confirmed Cases (lagged) | $\beta^{(d)}, \beta^{(u)}, \gamma, h$ |
| Deaths (lagged) | $\beta^{(d)}, \beta^{(u)}, \gamma, h$ |

This static covariate is encoded into the recovery rates ($\eta$), recovery ($\rho^{(I,d)}$, $\rho^{(I,u)}$, $\rho^{(H)}$, $\rho^{(C)}$, $\rho^{(V)}$) and death rates ($\kappa^{(I,d)}$, $\kappa^H$, $\kappa^C$, $\kappa^V$), at both the state- and county-level of geographic resolution.

**Econometrics**. We posit that an individual's economic status, as well as the proximity to other individuals in a region has an effect on the rates of infection, hospitalization and recovery. The proximity can be due to high population density in urban areas, or due to economic compulsions. The US census–available from census.gov and on BigQuery Public Datasets [8]–reports state- and county-level static data on population, population density, per capita income, poverty levels, households on public assistance (bigquery-public-data:census_bureau_acs.county_2018_5yr and bigquery-public-data:census_bureau_acs.county_2018_1yr). All of these measures affect transitions into the exposed and infected compartments ($\beta^{(d)}$, $\beta^{(u)}$), as well as the recovery rates ($\rho^{(I,d)}$, $\rho^{(I,u)}$, $\rho^{(H)}$, $\rho^{(C)}$, $\rho^{(V)}$) and death rates ($\kappa^{(I,d)}$, $\kappa^H$, $\kappa^C$, $\kappa^V$), at both the state- and county-level of geographic resolution. In addition, for the state-level model, it also influences the hospitalization rate $h$, ICU rate $c$ and ventilator rate $v$.

**Hospital Resource Availability**. We posit that when an epidemic of like COVID-19 strikes a community with such a rapid progression, local hospital resources can quickly become overwhelmed [9]. We use the BigQuery public dataset that comes from the Center for Medicare and Medicaid Services, a federal agency within the United States Department of Health and Human Services (bigquery-public-data:cms_medicare.hospital_general_info). These static covariates are encoded into the diagnosis rate ($\gamma$), recovery rates ($\rho^{(I,d)}$, $\rho^{(I,u)}$, $\rho^{(H)}$, $\rho^{(C)}$, $\rho^{(V)}$), re-infected rate ($\eta$) and death rate ($\kappa^{(I,d)}$, $\kappa^H$, $\kappa^C$, $\kappa^V$), at both the state- and county-level of geographic resolution.

**Confirmed Cases and Deaths**. Past confirmed case counts and deaths can have an effect on the current values of these quantities. We include these as temporal covariates. These have an effect on the average contact rates ($\beta^{(d)}$, $\beta^{(u)}$), the diagnosis rate ($\gamma$) and the hospitalization rate $h$.

# B  Comparisons to IHME model

In this section, we include the comparison of our model with Institute for Health Metrics and Evaluation (IHME) [10] model, which has been used by major US government organizations. IHME is based on curve-fitting considering the nonlinear mixing effects with intervention assumptions. Since the provided prediction dates are different for IHME model, we run separate comparisons for it in 12-day ahead forecasting setting (still using our 14-day forecasts). As Table 2 shows, our model significantly outperforms IHME, consistently across all phases of the disease.

Table 2: $\tau$-day ahead MAE for 12-day forecasting the number of deaths at state-level.

| Pred. horizon $\tau$ (days) | Pred. date | Ours | IHME |
|---|---|---|---|
| | 05/05/2020 | **128.6** | 146.8 |
| | 05/19/2020 | **56.1** | 116.1 |
| | 06/09/2020 | **43.4** | 60.5 |
| 12 | 06/23/2020 | **81.4** | 99.8 |
| | 08/25/2020 | **36.3** | 106.3 |
| | 09/22/2020 | **40.7** | 69.6 |

# C  Comparisons to Berkeley Yu model

In this section we include more comparisons of our model with the Berkeley Yu model [11] on more recent dates. As Table 3 shows, our model consistently outperforms Berkeley Yu model.

Table 3: $\tau$-day average MAE for 7-day forecasting the number of deaths at county-level.

| Pred. date | Ours | Berkeley-Yu |
|---|---|---|
| 2020/06/02 | **0.8** | 1.82 |
| 2020/06/09 | **1.02** | 1.79 |
| 2020/06/16 | **0.9** | 1.75 |
| 2020/07/21 | **1.11** | 2.11 |
| 2020/07/28 | **1.33** | 2.68 |
| 2020/08/11 | **1.36** | 2.33 |
| 2020/08/18 | **1.35** | 2.27 |

# D  Prediction intervals of confirmed cases

(a) CA        (b) FL        (c) TX

Figure 1: Prediction intervals of confirmed cases for 14-day forecasting. We use the 10-th and the 90-th quantile prediction as the the lower and the upper bound of prediction intervals, respectively.

# E  Impact of data quality

For fair comparison, we used the data available on the prediction date for model development (training and model selection), and we use the data from $\tau$ days later after the prediction date for evaluation. There are numerous data quality issues which are often corrected later. It is not unlikely to see that the number of confirmed or death cases for a particular day are significantly increased or decreased few weeks (and sometimes few months) later. Unfortunately, such data quality issues often have unpredictable patterns (due to human entry errors, reporting changes, or infrastructure issues) and can

be treated as input 'noise'. The accuracy of our model also suffers from them. To demonstrate the impact of data quality issues, we perform experiments by training and evaluating our model with the most recent version of the data (from October), on different cases (note that we still have the same time-series split for training, validation and test). Table 4 shows the significant difference in results and the potential of our model with better quality data.

Table 4: $\tau$-day average MAE for 14-day forecasting the number of deaths at state-level.

| Pred. horizon $\tau$ (days) | Pred. date | Ours (data version of pred. date) | Ours (recent data version) |
|---|---|---|---|
| 14 | 06/02/2020 | 32.8 | 29.6 |
| | 06/09/2020 | 28.8 | 23.2 |
| | 06/16/2020 | 31.4 | 20.9 |

# F Rate variable definitions

Table 5: Variables and the covariates that affect them. (doc.: documented, undoc.: undocumented)

| Variable | Description | Covariates |
|---|---|---|
| $\beta$ | **Average contacts** of doc. infected ($\beta^{(d)}$) / undoc. infected ($\beta^{(u)}$) | Mobility, Interventions, Density |
| $\eta$ | Re-infected rate | Census, Healthcare |
| $\alpha$ | Inverse latency period | - |
| $\gamma$ | Diagnosis rate | Census, Test info |
| $h$ | Hospitalization rate for infected | Census, Income, Healthcare |
| $c$ | ICU rate for hospitalized | |
| $v$ | Ventilator rate from ICU | |
| $\rho$ | **Recovery rate** for doc. infected ($\rho^{(I,d)}$), undoc. infected ($\rho^{(I,u)}$), hospitalized ($\rho^{(H)}$), ICU ($\rho^{(U)}$), ventilator ($\rho^{(V)}$) | |
| $\kappa$ | **Death rate** for doc. infected ($\kappa^{(I,d)}$), hospitalized ($\kappa^{(H)}$), ICU ($\kappa^{(C)}$), ventilator ($\kappa^{(V)}$) | |

# G Effective reproduction number

The effective reproduction number $R_e$ is the expected number of new infections arising directly from one infected individual in a population where all individuals are susceptible to infection [12]. For example, the $R_e$ for COVID-19 during the early stages of the pandemic in Wuhan, China has been estimated to be around 5.7 [13].

The Next-Generation Matrix [12] is a method to derive expressions for the $R_e$ from a given compartment model. The method involves first finding the disease-free equilibrium (DFE) of the model. The infected sub-system of the compartment model at DFE is identified and its corresponding differential equations are isolated. Then the inflow and outflow terms from each compartment in the sub-system are partitioned between two categories–(i) new infection causing events and (ii) all other flows between compartments.

Two matrices–the new infections matrix **F** and the transitions matrix **V**–are constructed from the inflow and outflow terms.

The DFE for our model is $[S, E, I^{(d)}, I^{(u)}, R^{(d)}, R^{(u)}, H, C, V, D] = [N, 0, 0, 0, 0, 0, 0, 0, 0, 0]$. We begin by isolating the infection subsystem as shown in Figure 2. All the individuals in these compartments $\overrightarrow{X} \equiv [E, I^{(d)}, I^{(u)}, H, C, V]$ are at some stage of the infection. The individuals in $\overrightarrow{Y} \equiv [S, R^{(d)}, R^{(u)}, D]$ are not infected. From the system of difference equations in Section **??**, the

Figure 2: Our compartment model with the infection compartments highlighted, and the variables from Table 5 indicated next to each transition.

differential equations for the infection subsystem reduces to:

$$
\begin{aligned}
\dot{E} &= (\beta^{(d)} \cdot I^{(d)} + \beta^{(u)} \cdot I^{(u)}) \cdot S/N_i - \alpha \cdot E \\
\dot{I^{(d)}} &= \gamma \cdot I^{(u)} - (\rho^{(I,d)} + \kappa^{(I,d)} + h) \cdot I^{(d)} \\
\dot{I^{(u)}} &= \alpha \cdot E - (\rho^{(I,u)} + \gamma) \cdot I^{(u)} \\
\dot{H} &= h \cdot I^{(d)} - \kappa^C \cdot (C - V) - \kappa^V \cdot V - (\kappa^{(H)} + \rho^{(H)}) \cdot (H - C) \\
\dot{C} &= c \cdot (H - C) - (\kappa^{(C)} + \rho^{(C)} + v) \cdot (C - V) + \kappa^{(V)} \cdot V \\
\dot{V} &= v \cdot (C - V) - (\kappa^{(V)} + \rho^{(V)}) \cdot V
\end{aligned}
\tag{1}
$$

At the DFE, the subsystem is:

$$
\begin{aligned}
\dot{E} &= (\beta^{(d)} \cdot I^{(d)} + \beta^{(u)} \cdot I^{(u)}) - \alpha \cdot E \\
\dot{I^{(d)}} &= \gamma \cdot I^{(u)} - (\rho^{(I,d)} + \kappa^{(I,d)} + h) \cdot I^{(d)} \\
\dot{I^{(u)}} &= \alpha \cdot E - (\rho^{(I,u)} + \gamma) \cdot I^{(u)} \\
\dot{H} &= h \cdot I^{(d)} - \kappa^C \cdot (C - V) - \kappa^V \cdot V - (\kappa^{(H)} + \rho^{(H)}) \cdot (H - C) \\
\dot{C} &= c \cdot (H - C) - (\kappa^{(C)} + \rho^{(C)} + v) \cdot (C - V) + \kappa^{(V)} \cdot V \\
\dot{V} &= v \cdot (C - V) - (\kappa^{(V)} + \rho^{(V)}) \cdot V
\end{aligned}
\tag{2}
$$

Examining the right-hand side of the system of equations 2, we see that it is of the form:

$$
\dot{\vec{X}} = \mathbf{M} \times \vec{X}
\tag{3}
$$

where $\mathbf{X} \equiv [E, I^{(d)}, I^{(u)}, H, C, V]$ and $\mathbf{M}$ is given by:

$$
\begin{bmatrix}
-\alpha & \beta^{(d)} & \beta^{(u)} & 0 & 0 & 0 \\
0 & -h - \kappa^{(I,d)} - \rho^{(I,d)} & \gamma & 0 & 0 & 0 \\
\alpha & 0 & -\gamma - \rho^{(I,u)} & 0 & 0 & 0 \\
0 & h & 0 & -\kappa^{(H)} - \rho^{(H)} & -\kappa^{(C)} + \kappa^{(H)} + \rho^{(H)} & \kappa^{(C)} - \kappa^{(V)} \\
0 & 0 & 0 & c & -c - v - \kappa^{(C)} - \rho^{(C)} & \kappa^{(C)} - \kappa^{(V)} + \rho^{(C)} + v \\
0 & 0 & 0 & 0 & v & -\kappa^{(V)} - \rho^{(V)} - v
\end{bmatrix}
\tag{4}
$$

Upon examination of Figure 2, we see that the only new-infection causing events are described by the rates $\beta^{(d)}$ and $\beta^{(u)}$. We define the new infections matrix $\mathbf{F}$ as:

$$\begin{bmatrix} 0 & \beta^{(d)} & \beta^{(u)} & 0 & 0 & 0 \\ 0 & 0 & 0 & 0 & 0 & 0 \\ 0 & 0 & 0 & 0 & 0 & 0 \\ 0 & 0 & 0 & 0 & 0 & 0 \\ 0 & 0 & 0 & 0 & 0 & 0 \\ 0 & 0 & 0 & 0 & 0 & 0 \end{bmatrix} \tag{5}$$

We calculate the transitions matrix $\mathbf{V} = -(\mathbf{M} - \mathbf{F})$ to be:

$$\begin{bmatrix} \alpha & 0 & 0 & 0 & 0 & 0 \\ 0 & h + \kappa^{(I,d)} + \rho^{(I,d)} & -\gamma & 0 & 0 & 0 \\ -\alpha & 0 & \gamma + \rho^{(I,u)} & 0 & 0 & 0 \\ 0 & -h & 0 & \kappa^{(H)} + \rho^{(H)} & \kappa^{(C)} - \kappa^{(H)} - \rho^{(H)} & -\kappa^{(C)} + \kappa^{(V)} \\ 0 & 0 & 0 & -c & c + v + \kappa^{(C)} + \rho^{(C)} & -\kappa^{(C)} + \kappa^{(V)} - \rho^{(C)} - v \\ 0 & 0 & 0 & 0 & -v & \kappa^{(V)} + \rho^{(V)} + v \end{bmatrix} \tag{6}$$

From $\mathbf{F}$ and $\mathbf{V}$ we get the Next-Generation Matrix $\mathbf{K} = \mathbf{F} \times \mathbf{V}^{-1}$:

$$\begin{bmatrix} \frac{\beta^{(d)}\gamma}{(\gamma + \rho^{(I,u)})(h + \kappa^{(I,d)} + \rho^{(I,d)})} + \frac{\beta^{(u)}}{\gamma + \rho^{(I,u)}} & \frac{\beta^{(d)}}{h + \kappa^{(I,d)} + \rho^{(I,d)}} & \frac{\beta^{(d)}\gamma}{(\gamma + \rho^{(I,u)})(h + \kappa^{(I,d)} + \rho^{(I,d)})} + \frac{\beta^{(u)}}{\gamma + \rho^{(I,u)}} & 0 & 0 & 0 \\ 0 & 0 & 0 & 0 & 0 & 0 \\ 0 & 0 & 0 & 0 & 0 & 0 \\ 0 & 0 & 0 & 0 & 0 & 0 \\ 0 & 0 & 0 & 0 & 0 & 0 \\ 0 & 0 & 0 & 0 & 0 & 0 \end{bmatrix} \tag{7}$$

Calculating the eigenvalues of $\mathbf{K}$ gives us 5 eigenvalues that are 0, and one non-zero eigenvalue, which is the spectral radius of $\mathbf{K}$. This is the effective reproduction number $R_e$:

$$R_0 = \frac{\beta^{(d)}\gamma + \beta^{(u)}(h + \kappa^{(I,d)} + \rho^{(I,d)})}{(\gamma + \rho^{(I,u)})(h + \kappa^{(I,d)} + \rho^{(I,d)})}. \tag{8}$$

## H  Training details

The start date of training is set to 1/21/2020. We assume that the compartmental equation regime start when the number of confirmed cases exceed 10 (before it, to avoid noise, we simple assign the initial values). We initialize the values as follows, where $\psi_{()} \sim U[0,1]$ denote random variables with uniform distribution: $\hat{E}_i[0] = \max(100\psi_{E_{i,1}}, 10\psi_{E_{i,2}}Q_i[0])$, $I_i^{(\hat{d})}[0] = Q_i[0]$, $I_i^{(\hat{u})}[0] = \max(100\psi_{E_{i,1}}, 10\psi_{E_{i,2}}Q_i[0])$, $R_i^{(\hat{d})}[0] = R[0]$, $R_i^{(\hat{u})}[0] = 5\psi_{R_i}R[0]$, $\hat{H}_i[0] = \mathbb{I}\{H[0]\}H[0] + 0.5\psi_{H_i}(1 - \mathbb{I}\{H[0]\})Q[0]$, $\hat{C}_i[0] = \mathbb{I}\{C[0]\}C[0] + 0.2\psi_{C_i}(1 - \mathbb{I}\{C[0]\})Q[0]$ and $\hat{V}_i[0] = \mathbb{I}\{V[0]\}V[0] + (1 - 0.5\psi_{V_i}\mathbb{I}\{V[0]\})Q[0]$. In general, our model is not too sensitive to random initialization of the initial values, and we just define wide ranges to enable exploration.

## I  Hospitalization forecasts

Fig. 3 exemplifies fitted hospitalization predictions for 8 states. Our model can provide robust and accurate forecasts consistently (e.g. in increasing, decreasing or plateauing trends), despite the fluctuations in the past observed data.

(a) NY      (b) NJ

(c) WA      (d) DC

(e) DE      (f) VT

(g) PA      (h) NH

Figure 3: Fitted hospitalization compartments for 8 states. The vertical line shows the prediction date. Our model can provide robust and accurate forecasts, despite the highly-noisy observed data.

# J State-level 14-day forecasts

We present the 14-day forecasts for all 50 US states.

Figure 4: Model performance on US states–Alabama to Minnesota.

Figure 5: Model performance on US states–Missouri to Washington.

# K County-level 14-day forecasts

We present the 14-day forecasts for selected US counties ordered by deaths from Covid-19.

Figure 6: Model performance on selected US counties ordered by deaths.

## L  Potential limitations

In this section, we list the potential limitations and failure cases of our model, to guide those who may use the techniques to build forecasting systems that may effect public health decisions:

- **Ground-truth data issues:** We are using different case counts data to supervise model training. It has been noted that the ground truth case counts might not be completely accurate for various reasons, such as the practices to obtain case counts varying across locations [14, 15]. We have weighted optimization to balance supervision from different signals, e.g. we have higher weight on the supervision from the death case count because it is known to be more accurate. Yet, the case counts data quality may also vary across locations and may affect our model's performance.

- **Failure to capture very rapid trend changes:** When the case count curves suddenly becomes very flat or very sharp, our model can fail to capture such dynamics. Some of such trends occur due to modifications in reporting practices, and some due to other factors that are not captured by the covariates we use. We believe more optimal temporal encoding approaches and integration of additional time-varying covariates may further mitigate this.

- **Using equal weight for all locations:** Our goal is to define a nation-wide metric that represents all individuals. We do not apply hand-tuned weighting for different locations, although it would be trivial with our framework. When equal weights are considered, the locations with the high case counts dominate the learning, which is often desired, but if any application requires a different emphasis mechanism, such as more accuracy for locations specifically with higher average age etc., the coefficient of the constituent locations' loss terms can be re-weighted.

- **Having symmetry in loss:** Under- vs. over-prediction have different implications on public policy, socioeconomic dynamics and public health. Our framework allows penalizing them in an asymmetric way, and we have tried different weights but we could not obtain consistent improvements when the overall accuracy is considered. Instead of overall accuracy, if an application needs to focus on under- or over-prediction specifically, the model could be retrained for improved performance.

- **Performance differences among sub-groups:** As COVID-19 is affecting certain sub-groups more than others, the case counts are not uniformly distributed among the entire population. As absolute errors tend to be higher for greater case counts, there could be performance differences among different sub-groups. We do not observe our model to exacerbate the inherit case count differences (e.g. for racial and ethnic subgroups).

- **Overfitting to the past:** Especially in very early phases of the disease, our model may suffer from overfitting in some cases, as the past observations may not have sufficient information content for all the dynamics of the future. We have various mitigation mechanisms to prevent overfitting, but it is impossible to completely get rid of it. We overall observe improved performance relative to the benchmarks with more training data.

- **Prediction intervals and uncertainty:** Our approach to obtain prediction intervals is not based on Bayesian approaches [16, 17] per se (we do not estimate the posteriors of the parameters). We adapt quantile regression to obtain prediction intervals, and we cannot decouple the aleatoric (statistical) vs. epistemic (systematic) uncertainty inherent in the data and the model, while training. An ideal forecasting model should be able to decouple those and while providing accurate (point) forecasts, it should be able to tell the range of scenarios accurately (in a well-calibrated way). We leave such Bayesian approaches to improve our base ideas to future work.