[Reviews · NeurIPS 2020]

Review 1

Summary and Contributions: Combines compartmental disease modeling with machine learning for COVID-19 forecasting. It surpasses state-of-the-art methods by some well motivated additions to existing models. It also incorporates a level of interpretability. Contributions include an extension to the standard SEIR model with additional compartments, time-varying encoding of the covariates, and learning mechanisms to improve generalization while learning from limited training data.

Strengths: Systematically integrated additional covariates into the compartmental model of SEIR. Allows covariates to be time-varying. Appendix is very detailed explaining variables and each data source so the results can be reproduced. Performance exceeds all baselines. The paper also includes county level forecasts. The additional results provided in the appendix further support the paper's claims. The demonstrated explainable insights also provide support to the interpretability claim.

Weaknesses: Does not provide code and a lot of relevant information is pushed to appendix. Most of this information is required for reproducing the results. Given the constraints of paper length, I consider this acceptable. Line 127: Table 2 does not contain the variables mentioned here. I believe this refers to Table 2 in Appendix?

Correctness: Yes both are correct. The method is developed incrementally and is easy to follow.

Clarity: Yes

Relation to Prior Work: Yes, it discusses prior work in good detail and clearly shows how it defers from them.

Reproducibility: Yes

Additional Feedback: Overall, there are enough contributions and improvement in results from existing approaches. The interpretability claim is also justified through multiple demonstrations. The appendix is very detailed and when seen in conjunction with the paper, it is easy to follow the method. All data is pulled from open sources. Would still prefer the code to be released. Having read other papers of COVID-19 forecasting, this paper appears to me as well thought out with enough important contributions to be very useful. Finally, the authors feedback was great in addressing comments.


Review 2

Summary and Contributions: This paper proposes a novel approach that modeling COVID-19 progression. The authors extend the standard SEIR model with a newly designed compartment for undocumented cases and hospital resource usage and also incorporate understandable co-variates such as mobility indices into the model. During training, the authors combine several techniques to overcome the overfitting problem and improve generalization. Using the current public US COVID-19 dataset, authors compared the performance with a number of current public benchmark approaches.

Strengths: Compare to other epidemic forecasting, the major challenge in current COVID-19 pandemic forecasting is that there are many potential sources of data, but their causal impact on the disease is unclear and the progression of the disease influences will largely influence the public policy and individuals’ public behaviors and vice versa. It's important to extract useful features from these covariates and design a proper system to incorporate them into the model. Thus, the major contribution of this paper can be summarized as: 1. The proposed compartmental model is novel and carefully tailored for COIVD-19. Based on the standard SEIR model, authors introduce compartments such as undocumented infected and recovered cases, hospitalized/ICU and ventilator to the model. They also give proper assumptions such as partial immunity based on the latest medical research. 2. Authors select many important covariates that could have an impact on the model compartments such as mobility, hospital Resource availability, and use an encoder to incorporate these covariates into the model. In addition, in this design, the proposed model can provide explanatory insights, for example, the mobility index has a contribution to the infectious rate while the school close has a negative weight.

Weaknesses: 1. Related Work. In the related work section, the authors summarize some related models for infectious diseases and address their weaknesses. However, it seems none of them are used as baseline models for comparison in the experiment section. Instead, the authors present the results of five top-performing models designed for COVID-19. It would be better if the authors can give a summary of these COVID-19 models to address their weakness and point out the major improvement of their method compared with these ones. 2. Experiment: a. Part of the ablation studies is unclear to me. The author claims the extra compartment has significant benefits for the prediction. However, Table 4 does not directly show this benefit. If we compare the first row and second row, the covariates encoder seems to offer the largest benefit, but the author does not mention this. It would be better if the author makes a clearer table, and show different results based on a different combination of the techniques they applied. b. The author compares their model with 5 top-performance COVID-19 forecasting models and claims their method outperforms the next best model by a large margin. However, in Table 3 in Appendix H., when the prediction horizon is 5 days, although the author bolds their results, their model cannot beat YYG in 3/5 dates. In addition, the comparison is unfair, the proposed model uses an additional 16 covariates data, while others are different. I would suggest the author provide a deeper discussion between their model and others instead of citing some number, especially the YYG model which is also based on the SEIR and have close performance.

Correctness: As mentioned before, the comparison seems unfair, the author would justify, even with additional encoder of co-variate data, why their model can not beat YYG when prediction horizon is small.

Clarity: The paper is clearly written and not hard to follow. However as stated before, a proper related work and more insight discussion are desirable.

Relation to Prior Work: Not completely, as mentioned before, I would discuss the major differences between their models and other COIVD-19 forecasting methods they compared in the experiment part.

Reproducibility: No

Additional Feedback: By updating "related work", adding new "ablation study" and ''model comparison" sections, authors have addressed most of my concerns satisfactorily. Thus I would like to increase my overall score and recommend acceptance of this paper.


Review 3

Summary and Contributions: The paper proposes a new compartmental model by improving an well-established model for contagious viral diseases, accounting for undocumented cases, hospitalization, icu stays and necessity for ventilator use.

Strengths: This work improved an well-established model to incorporate factors that were identified as important in the current global pandemic. COVID-19 is currently the most relevant topic globally and improving upon a traditional model and also providing tools that utilize other techniques to complement a compartmental model fits the purpose of the conference very well.

Weaknesses: The authors did elaborate well on how their model could be actually used by policy makers. The work would be highly enriched if they introduced why their choices improve decision making as a reasoning behind their modeling.

Correctness: There were no identified errors in their claims and methodologies.

Clarity: The paper is well written and all the methods all well described. The Supplemental material is valuable and provides useful insights into their development methods.

Relation to Prior Work: The authors discussed previous work as well as provided clarity as to how their work differed from the other similar models.

Reproducibility: No

Additional Feedback: - The development of forecasting tools for covid-19 has been a popular topic. The authors should tone down their claims that their model outperforms the next best model as things are changing quickly and literature on this has vastly increased over short periods. - The use of static rates is the reason SEIR has been popular among policy makers. For example, it allows them to make quickly change the social distancing measures based on the trend of the forecast given a mobility rate. The new compartments added by the authors can allow them to make decisions regarding other aspects such as availability and relocation of ICU beds and ventilators. While introducing a trainable model to create time-sensitive variables does give insight to what covariates might be affecting the current trends, it will limit the capability to simulate other scenarios as the model will fail to incorporate major changes in policy, which is partially due to collinearity and also due to the fact that their model does not account for how much each timestep influences the prediction interval. This could be observed in the paper, but the authors did not state this as a limitation. Minor fixes: - There is a typo in the figure 2, the recovery rate for the icu recovery p(c) is written as p(u).

[Author Response · NeurIPS 2020]

1 We would like to thank all the Reviewers for their valuable comments that have helped improve our submission.

2 **Reproducibility [R1]:** We have put special attention into including all details (initialization, optimization algorithms, parameter values) for reproducibility, and we are working on open-sourcing our code so others can use it conveniently.

4 **Related work [R2]:** We agree that our Related Work would benefit from descriptions of comparison models. We briefly discuss them here and will expand the explanations in the paper: **(i) YYG** is an SEIR model with learnable parameters and accounts for reopenings. The parameters are fit using hyperparameter optimization. Unlike ours, YYG uses fixed (time-invariant) rates as SEIR parameters and is limited to modeling standard SEIR compartments. It does not have a systematic mechanism to integrate additional covariates and it cannot benefit from cross-location information sharing. **(ii) IHME** is based on fitting a curve to model the non-linear mixing effects. Unlike ours or others based on compartmental modeling, it does not explicitly model the transitions between the compartments and thus cannot reflect well important inductive biases from epidemiology. **(iii) LANL** is based on statistical-dynamical growth modeling for the underlying numbers of susceptible and infected cases. Unlike ours, it does not model all the available compartments and it does not have any mechanism to learn from covariates. **(iv) UMass** is a Bayesian compartmental model separately fit to each location considering observation noise and a detection rate. Unlike ours, it does not have any mechanism to share information across locations and it does not utilize information extracted from covariates.

16 **Ablation study [R2]:** We've added extra ablation studies to clarify the contributions of different model parts. Due to space limit, we only show results for 3 dates in Table 1. Our model with additional compartments outperforms standard SEIR models (only with $S$, $E$, $I^{(d)}$ and $R^{(d)}$ compartments), with or without encoder (1st vs. 2nd and 3rd vs. 4th rows). Encoders bring significant performance gains, for both cases. Regularization also helps especially during phases of rapid trend change (e.g. 05/25 and 06/01).

Table 1: 14-day forecasting RMSE for state-level deaths.

| Models / Prediction date | 05/25 | 06/01 | 06/08 |
|---|---|---|---|
| Standard SEIR compartments (*w/o* encoder) | 219.8 | 190.8 | 127.1 |
| Standard SEIR compartments (*with* encoder) | 116.4 | 75.4 | 81.2 |
| Our model (*w/o* encoder) | 128.5 | 77.0 | 50.0 |
| **Our model** | **57.6** | **41.1** | 42.0 |
| **More ablation cases** | | | |
| Our model *w/o* fine-tuning | 179.9 | 143.0 | 122.2 |
| Our model *w/o* partial teacher forcing | 1311.6 | 3141.2 | 2163.6 |
| Our model *w/o* regularization | 130.8 | 132.5 | **40.3** |

28 **Model comparison [R2]:** We will correct the bold fonts in Appendix. We have been running our model on new dates since our submission and we include those results here to help increase the confidence in our results. Overall, Table 2 shows that *our model consistently outperforms YYG (as well as others that we cannot include due to space limits) by a large margin* in most cases, and indeed, the gap is even larger margin for these recent dates. We attribute this to learning from more training data – by using covariate encoders our model takes advantage of the increasing training data in ways other methods cannot.

Table 2: RMSE for forecasting the number of deaths at state level with different prediction dates and forecasting horizons.

| Start date | Predict 5 days | | Predict 7 days | | Predict 14 days | |
|---|---|---|---|---|---|---|
| | Ours | YYG | Ours | YYG | Ours | YYG |
| 05/24/2020 | **51.4** | 142.4 | **41.5** | 143.9 | **76.7** | 158.3 |
| 05/25/2020 | **36.6** | 138.2 | **31.3** | 140.6 | **61.5** | 150.8 |
| 05/31/2020 | **37.7** | 152.9 | **37.6** | 154.4 | **53.2** | 163.1 |
| 06/01/2020 | **28.5** | 153.4 | **34.7** | 155.2 | **44.9** | 161.7 |
| 06/07/2020 | **33.8** | 157.2 | **32.8** | 159.1 | **50.0** | 165.3 |
| 06/08/2020 | **27.5** | 150.8 | **30.3** | 152.1 | **42.5** | 157.6 |
| 06/15/2020 | **39.6** | 151.5 | **31.4** | 153.3 | **139.6** | 211.0 |
| 06/22/2020 | **184.3** | 225.3 | **200.8** | 270.1 | **224.9** | 296.3 |
| 06/29/2020 | **90.8** | 158.4 | **110.1** | 160.1 | **96.6** | 168.3 |

40 **Using covariates [R2, R4]:** We agree that using more covariates is advantageous as they contain additional information. Systematic integration of covariates is under-explored in existing epidemiology literature and is one of our major contributions. We convey their value by demonstrating large accuracy gains. We also show that our model outperforms other methods that use covariates such as IHME. We will clarify our contribution as efficient information extraction from extra covariates for epidemiological modeling. We will also add the limitations mentioned by R4.

46 **Policy making use cases [R4]:** We will include more explanations on how forecasts can guide better policy decisions. In summary, the benefits for public health are more optimal (i) resource allocation and (ii) non-pharmaceutical interventions. For (i), healthcare organizations rely on accurate forecasts to reallocate health personnel, protective equipment, ventilators and treatment drugs based on the expected severity of COVID-19. E.g. if one county has higher hospitalization forecasts compared to its neighbor, the resources can be shifted to that county to help better address the outbreak. For (ii), local governments can make better-informed decisions on gathering bans and school/business/restaurant closures, as well as improve their messaging to the public based on the expected severity of COVID-19. *Our forecasts already being actively used by several large healthcare organizations and governments, and are publicly available – we will add some notes on these later (we cannot disclose here due to NeurIPS anonymity principles).*

55 **Toning down our claims [R4]:** As explained to R2, we have been running benchmarks continuously since our submission and we do seem to be consistently outperforming other models by a large margin. That being said, we appreciate this comment and we agree that we should not make strong claims based on the limited data given the non-stationary environment. We will revise all our claims to tone them down.

59 **Typos [R1,R4]:** Line 127 refers to the Table 2 in Appendix. Thanks for all found typos, we will correct them.

[Meta-Review · NeurIPS 2020]

Thanks for your paper submission to NeurIPS. Three knowledgeable reviewers support acceptance, particularly due to the proposed model novelty, performance, interpretability, and relatively good reproducibility. I agree with these strengths and overall paper contribution and must accept. For the final version, please strongly consider enhancing the paper with (1) Code: making the code available will help with further research and reproducibility (2) Related work: it would be a great addition to the paper if the five top-performing models in the experiment section were explained in better detail in the related work section (see R2). (3) Limitations: address R3 feedback regarding acknowledging limitations. Thank you.